# COMBATING HIDDEN VULNERABILITIES IN COMPUTER VISION TASKS

## ABSTRACT

Backdoor attacks are among the most prominent security threats to deep learning models. Traditional backdoors leverage static trigger patterns, such as a red square patch. They can be removed by existing defense techniques. However, recent backdoor attacks use semantic features as the trigger. Existing techniques largely fall short when facing such backdoors. In this paper, we propose a novel backdoor mitigation technique, MARTINI, that effectively mitigates various backdoors. It features a specially designed trigger reverse-engineering method for constructing backdoor samples that have a similar attack effect as the injected backdoor across a spectrum of attacks. Using the samples derived from MARTINI, paired with the correct labels, in training can remove injected backdoor effects in deep learning models. Our evaluation on 14 types of backdoor attacks in image classification shows that MARTINI can reduce the attack success rate (ASR) from 96.56% to 5.17% on average, outperforming 12 state-of-the-art backdoor removal approaches, which at best reduce the ASR to 26.56%. It can also mitigate backdoors in self-supervised learning and object detection.

## 1 INTRODUCTION

Deep learning is widely used in various critical applications, such as autonomous driving (Cao et al., 2021), face recognition (Parkhi et al.), and disease diagnosis (Li et al., 2014). Despite their near-perfect performance on these tasks, it is not difficult for attackers to manipulate the behavior of deep learning systems and induce attack-intended output. For example, backdoor vulnerabilities in deep neural networks can be triggered by adding *backdoor triggers* to inputs, causing misclassification to a *target label* (Gu et al., 2019; Liu et al., 2018b).

A common backdoor employs static trigger patterns, such as a small square patch with a solid color (Gu et al., 2019). These trigger patterns are easy to construct and can be easily learned by deep learning models during training, as they are simple features. However, their distinctive features make them easily distinguishable from benign features of the original learning task. many defense techniques are able to successfully remove backdoor effects from attacked models (Wu & Wang, 2021; Li et al., 2023; 2021a; Zhu et al., 2023a).

However, a clear separation between backdoor features and clean-task features is not a necessary condition for a successful backdoor attack. There is a body of semantic backdoor attacks that modify the entire input, making changes either closely relevant to the main content or visually invisible (Chen et al., 2017; Barni et al., 2019). Different perturbations are applied to different inputs based on the input content. For example, the Deep Feature Space Trojan (DFST) (Cheng et al., 2021) leverages a generative adversarial network (GAN) to inject a certain style (e.g., sunrise color style) into the input. WaNet (Nguyen & Tran, 2021) leverages elastic image warping to deform an image through distortion transformation (e.g., distorting straight lines). The unique nature of these attacks renders most existing solutions less effective.

In this paper, we propose a novel backdoor mitigation technique through trigger reverse-engineering and model hardening. Specifically, we introduce MARTINI, which (re)constructs backdoor samples from backdoored models that closely resemble the injected backdoor effects. The constructed samples, paired with correct labels, are subsequently utilized for training the potentially backdoored model. MARTINI can model the trigger function for a wide range of backdoors. It manipulates abstract features instead of raw pixels to transform the input and achieve the backdoor effect,

inducing targeted misclassification. Specifically, given the feature representation of a clean input (from a pre-trained encoder), `MARTINI` leverages a unique transformation layer to mutate the feature representation. Its novel design allows us to express a wide range of attacks through feature mutation (as formally explained in Section 4.2). We use gradient descent to update the transformation layer so that the feature perturbation, as defined by the layer, can change the model's classification to a target label. As a result, the trained transformation layer captures the backdoor vulnerability in the model, if any.

Our contributions are summarized as follows.

- We propose a novel and effective technique, `MARTINI`, for mitigating hidden backdoors intentionally injected by adversaries.
- We develop a general formulation of backdoor trigger functions using a novel transformation layer. This design allows us to model a wide range of existing attacks, and consequently, training with our generated backdoor samples can improve model robustness against those attacks.
- We evaluate `MARTINI` on 14 types of backdoor attacks in image classification tasks. Our method can reduce the attack success rate (ASR) from 96.56% to 5.17% on average, surpassing 12 state-of-the-art backdoor removal techniques, which at best reduce the ASR to 26.56%. We also conduct experiments on two additional computer vision tasks: self-supervised learning and object detection. For self-supervised learning, `MARTINI` successfully reduces the ASR from 97.17% to 8.99%, whereas the best baseline only reduces it to 29.44%. For object detection, `MARTINI` reduces the ASR from 97.06% to 5.79%, significantly lower than the baselines, which achieve 34.56% at best. We further apply `MARTINI` to natural language processing models and demonstrate its generalizability in mitigating backdoors in other domains. Additionally, our adaptive attacks against `MARTINI` validate its robustness against knowledgeable adversaries.

## 2 THREAT MODEL

**Attack Goal and Capabilities.** The attacker aims to inject backdoors into deep learning models so that any input with the attacker-chosen trigger will be misclassified to a target output. The attacker can utilize a variety of backdoors that either use a patch-like pattern, such as BadNets (Gu et al., 2019) and Dynamic attack (Salem et al.), or apply semantically relevant perturbations that cover almost the entire input area, such as DFST (Cheng et al., 2021), WaNet (Nguyen & Tran, 2021), and Filter attack (Liu et al., 2019). The backdoor can flip any sample from any class to a target class, known as a *universal backdoor* (Gu et al., 2019; Chen et al., 2017; Liu et al., 2020), or flip samples from a particular victim class to a target class, known as a *label-specific backdoor* (Wang et al., 2019; Salem et al.; Liu et al., 2019). The attacker can inject backdoors into models either by poisoning the training data or by directly constructing a trojaned model to be published on online platforms (e.g., Huggingface).

**Defense Goal and Capabilities.** Given a pre-trained model, the goal of the defense is to mitigate potential backdoors in the model without knowing the backdoor types. The defense process should not affect the model's normal functionality, e.g., without sacrificing much accuracy. The defender has access to a subset of the clean training dataset (5%) (Li et al., 2021a; Tao et al., 2022a). The defender has full control over modifying the model, such as updating model weight parameters, removing neurons, etc. The updated model by the defender is not accessible to attackers.

## 3 MOTIVATION

Recent backdoor attacks adopt triggers that are more interconnected with the main task of learning models. Figure 1 shows a few example attacks. The first row presents clean inputs, and the second row shows these inputs modified by backdoor triggers. The differences between the images in the first two rows are given in the third row, illustrating how these backdoor attacks perturb inputs. Observe that all the pixels in the input are altered. Specifically, the DFST attack (Cheng et al., 2021) (1st column) applies a sunrise color style to the mountain image, disguising it as a natural scene.

The Blend attack (Chen et al., 2017) (2nd column) adds salt-and-pepper-like noise to the image, making it appear as a dog with colorful fur. The third case (SIG (Barni et al., 2019)) looks like a picture taken behind a fence. The changes introduced by WaNet (Nguyen & Tran, 2021) (4th column) are almost invisible as it only deforms an image through distortion transformations, such as twisting the outlines of objects. The last case by the Filter attack (Liu et al., 2019) (5th column) resembles an antique picture taken in the last century.

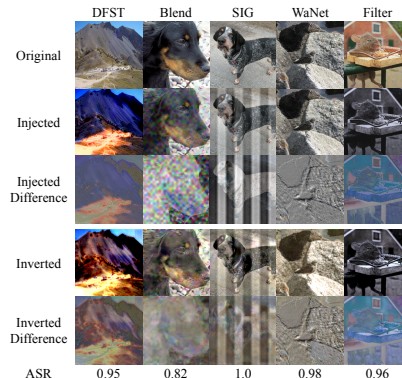

Figure 1: Examples of backdoor attacks (in the top block) and inverted trigger forms (in the bottom block)

All these backdoor attacks perturb inputs in a way that makes the trigger-inserted samples very similar to natural inputs. The additive features introduced by the triggers are relevant to the main content, making them not easily separable from the original learning task. The unique nature of these attacks makes mitigating backdoors extremely challenging and renders most existing solutions less effective.

### 3.1 LIMITATIONS OF EXISTING TECHNIQUES

**Relying on Neuron Isolation.** As backdoor attacks have a different goal from the original learning task, certain parts of backdoored models might be used specifically for achieving the attack goal, i.e., causing the target misclassification when the trigger is presented. Based on this assumption, several existing defense techniques aim to identify and remove the neurons compromised by backdoor attacks (Liu et al., 2018a; Li et al., 2023). For example, ANP (Wu & Wang, 2021) first identifies compromised neurons whose weight values are exceptionally sensitive and then prunes these neurons. RNP (Li et al., 2023) removes neurons that cause a large loss increase when reversing the original training objective, i.e., enlarging (instead of decreasing) the classification loss on clean data.

However, backdoor attacks may not activate a particular set of neurons. For instance, the DFST attack was designed specifically to reduce the identification of possible compromised neurons. During the attack, it applies a technique (Liu et al., 2019) to locate compromised neurons and then mitigates their presence, a process called *detoxification*. This process can significantly degrade the defense performance of existing techniques relying on neuron isolation. For example, after three rounds of detoxification, ANP can only reduce the attack success rate of DFST to 90.11%, with more than a 7% clean accuracy drop, which nearly fails to mitigate backdoors. This type of backdoor defense, which relies on neuron isolation, is less effective when backdoor-related neurons are interleaved with those for the original tasks, especially against attacks like those in Figure 1.

**Assuming Feature Separation.** Backdoor behavior may not be distinctive from clean-task behavior at the neuron level, as discussed earlier. It could be distributed across all layers and neurons in deep learning models. However, backdoor features could still be quite different from those of clean data. Another line of defenses assumes that by focusing on the original clean task, the model can "forget" the backdoor behavior (Zhu et al., 2023b; Min et al., 2024; Zeng et al., 2021). For example, FT-SAM (Zhu et al., 2023a) utilizes an optimization method called *sharpness-aware minimization* for fine-tuning the backdoored model on clean data, aiming to instruct the model to focus on the clean task. NAD (Li et al., 2021a) applies knowledge distillation techniques to extract a clean student model from the backdoored teacher model. The distillation process is also guided by the clean data to ensure the main task performance.

The assumption of feature separation however is not a necessary condition for backdoor attacks. A very recent attack, COMBAT (Huynh et al., 2024), only uses low-frequency components as the trigger, which are the channels where normal clean features lie. Defense methods that assume feature separation of backdoor attacks are less effective against such an attack. For instance, FT-SAM and NAD can only reduce the attack success rate of COMBAT to 67.33% and 42.65%, respectively.

**Less Powerful Trigger Modeling.** Trigger inversion is an approach that reverse-engineers the injected trigger from backdoor attacks (Tao et al., 2022b; Wang et al., 2022). This approach requires modeling the trigger function to find a set of parameters resembling the injected backdoor. For example, NC (Wang et al., 2019) uses two input-size matrices to denote which pixel and how much of the

pixel value should be changed. The inverted trigger can then be used to unlearn the backdoor effect by adding it to clean inputs and training them with the correct labels. ABS (Liu et al., 2019) leverages a simple convolutional kernel to model possible changes by attacks, such as those in Figure 1.

These existing inversion techniques cover only a very limited number of possible attacks. NC-like methods are mainly designed for patch-type backdoors and are not capable of modeling style-based attacks such as DFST (Cheng et al., 2021). The formulation of ABS is also not general enough for modeling various attacks. For example, when using ABS to reverse-engineer the trigger from the SIG attack (Barni et al., 2019), the inverted trigger achieves only a 39% attack success rate, compared to the injected trigger's 93%, indicating the inverted trigger does not resemble the injected one. Therefore, using this trigger in unlearning cannot mitigate the backdoor. In fact, the attack success rate remains 85.29% after applying ABS to purify the backdoored model.

# 4 METHODOLOGY

## 4.1 DEFENSE OVERVIEW

The workflow of our backdoor mitigation method, MARTINI, is presented in Figure 2. It consists of three steps: (1) decoder construction, (2) trigger reverse-engineering, and (3) backdoor mitigation.

In the first step, the features extracted from a pretrained encoder are fed to a decoder. The decoded image is compared to its original counterpart. The difference between these two is utilized as loss (for minimization) to update the decoder's weights. Only clean images (from the encoder's dataset, i.e., ImageNet) are used during decoder training. Once the training converges, the decoder is able to faithfully project abstract features to the input space.

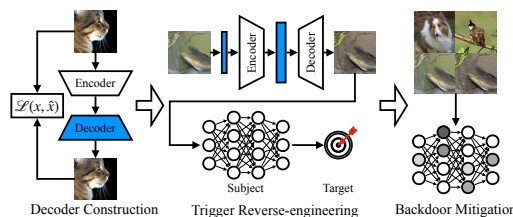

Figure 2: Overview of MARTINI

In the second step, MARTINI aims to transform a set of benign inputs into backdoor samples that can induce targeted misclassification. Specifically, given a set of clean inputs (from the victim model's dataset), it first normalizes the input values using a normalization layer such that different input samples have the same value distribution (i.e., mean and standard deviation). The feature representations from these normalized inputs are modified by our proposed transformation layer (blue rectangle in the middle), serving as the backdoor function. That is, **the transformation layer can inject backdoor features into the original feature representations**. Once decoded by the decoder, they can induce misclassifications on the victim model to the target label, having the same attack effect as the injected backdoor.

In the last step, the generated backdoor samples by MARTINI together with clean inputs are then used for training the victim model. It is an iterative procedure for steps 2 and 3. That is, for each training iteration, a few clean samples are chosen to generate backdoor samples with respect to the current state of the model as discussed in step 2. MARTINI **searches for different parameters of the transformation layer maximizing the attack ability that denotes various injected backdoors.** The generated samples paired with the correct labels are then used to update the victim model's weights to remove those backdoors. The training terminates when the model converges.

## 4.2 TRIGGER REVERSE-ENGINEERING AND BACKDOOR GENERATION

The goal of having a generic trigger function is to create a universal way of modeling various backdoor attacks. Our intuition is that in backdoor attacks, especially for semantic backdoors shown in Figure 1, the perturbation for a particular pixel $x_{i,j}$, denoted as $p_{i,j}$, is dependent on the original pixel values in its neighboring area. That is, $p_{i,j} = g(x_{i-\epsilon,j-\epsilon}, \ldots, x_{i+\epsilon,j+\epsilon})$. However, the function $g$ and the bound $\epsilon$ vary significantly from attack to attack, and even by different locations $i$ and $j$[1]. These perturbations induce feature space variations that can be approximated by a transformation layer (e.g., a convolutional layer). *Different attacks are essentially different sets of parameters of the transformation layer.* Our case study later in this section and empirical results in Section 5.1 support

---

[1]Patch-type backdoors are typically dependent primarily on locations, not on neighboring pixel values.

this argument. In the following, we first elaborate on the overall design of backdoor generation and then discuss each component in detail.

Figure 3 illustrates the procedure of our backdoor generation. It is carried out on a set of inputs. Here, we use one single input for discussion simplicity. Given an input $x \in \mathbb{R}^{C \times W \times H}$ ($C, W, H$ denote channel, width, and height, respectively), we first apply a normalization layer $\Gamma$ to obtain a normalized input $x'$. Input $x'$ is then fed to a pre-trained encoder $f$ (not the victim model) for obtaining the feature representation $a'$. Our backdoor transforma-

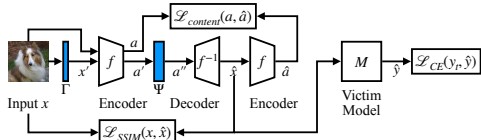

Figure 3: Procedure of generating backdoor samples from clean inputs

tion layer $\Psi$ adversarially modifies the representation $a'$ and produces an altered representation $a''$. The decoder $f^{-1}$ takes in $a''$ and generates a backdoor sample $\hat{x}$. We use the SSIM score (Wang et al.) as the loss function to constrain the difference between the backdoor sample $\hat{x}$ and the original input $x$. The backdoor sample $\hat{x}$ is also fed to the encoder $f$ to obtain its feature representation $\hat{a}$, which is used to compare with the original representation $a$ from the input $x$. We use the mean squared error as the content loss to bound the difference between $a$ and $\hat{a}$. To achieve the backdoor effect that can induce misclassification, the decoded backdoor sample $\hat{x}$ is passed to the victim model $M$ to obtain the prediction $\hat{y}$. The cross entropy loss is utilized to make sure the prediction $\hat{y}$ is the same as the target label $y_t$. **The normalization layer $\Gamma$ and the transformation layer $\Psi$ are optimized** during the backdoor generation. They serve as the trigger function to transform benign inputs to backdoor samples. We elaborate the details of $\Gamma$ and $\Psi$ as well as the loss terms in the following.

**Normalization Layer.** Different input samples may have distinct value distributions on each channel (i.e., R, G, B channels). For instance, an input $x_0$ may have all small values (e.g., 10) on the R channel, but another input $x_1$ has all large values (e.g., 200). A slightly larger transformation on $x_0$ is reasonable but can cause the change on $x_1$ out of the valid range (i.e., 255). It is hard for the optimization to find a valid solution for $x_1$ as it can be quickly out of the range. To facility an easier optimization process, a normalization layer $\Gamma$ is introduced in our backdoor generation. It is applied on the inputs to reduce the covariate shift on each channel. In other words, different inputs will have the same mean and standard deviation of pixel values for a particular channel (e.g., the R channel). Each channel has its own statistics. The normalization layer $\Gamma$ is defined as follows.

$$x' = \Gamma(x) = (x - \mu_x)/\sigma_x \cdot \sigma_b + \mu_b, \tag{1}$$

where $\mu_x$ and $\sigma_x$ are the mean and standard deviation of input $x$ along the width and height dimensions. That is, we have one mean value and one standard deviation value for each channel (e.g., $\mu_x \in \mathbb{R}^C$). Parameters $\mu_b$ and $\sigma_b$ are the normalization scaling variables in the same shape of $\mu_x$ and $\sigma_x$. Note that variables $\mu_b$ and $\sigma_b$ are the same for all the samples and will be optimized during our backdoor generation.

**Transformation Layer.** A backdoor sample derived from a clean input has a different internal feature representation as that of its clean counterpart. Since the exact injected backdoor a model has is unknown beforehand, we propose a transformation layer $\Psi$ to mutate the feature representation of the clean input, aiming to produce a feature representation that could cause misclassification like some backdoor sample. The transformation layer shall be general, allowing us to model a large spectrum of possible backdoors. As backdoors can alter all the pixels in the input, the changes can be diverse for different input regions.

Figure 4 presents an example. The first column shows two clean input images. The second column shows the injected backdoor samples that are transformed from clean inputs using a Toaster filter. Observe that the injected backdoor samples have dark orange color in the middle and lighter color for the surrounding areas. A straightforward design of the transformation layer is to use a traditional convolutional layer to transform the clean feature representation. The convolution operation denotes a uniform transformation, where all the values on a feature map is computed by a same kernel. However, this is undesirable for expressing the backdoor discussed above. The

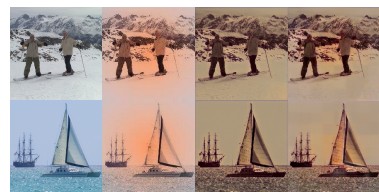

Input   Injected   Generated   Gen. w/ Region

Figure 4: Example of regional transformation

third column in Figure 4 denotes the generated samples by using a traditional convolutional layer. Observe that the color changes are uniform for different regions, failing to produce the orange color

region in the middle. We hence propose to divide a feature map into a set of regions and apply different convolutional kernels on different regions. We call it *regional transformation*. While the details are discussed later in this section, the last column in Figure 4 presents the results of using regional transformation for generating backdoor samples. Observe that comparing to the images in the third column, the regional transformation is able to produce the orange color in the middle and lighter color in the surrounding areas. Note that the Toaster filter is only one of the cases where backdoors manipulate different regions of the input using different transformations. *MARTINI is not restricted to the particular style of Toaster filter.* As shown in the bottom of Figure 1, our method can faithfully model a wide range of backdoor attacks with high ASR (shown in the last row).

We formally define the regional transformation in the following. Assume the input feature representation $a' \in \mathbb{R}^{C' \times W' \times H'}$ (features before transformation), and a set of convolutional kernels (i.e., weight parameters) $\boldsymbol{U} \in \mathbb{R}^{z \times z \times C' \times C' \times m \times m}$, where $z \times z$ is the number of convolutional kernels (one for each region in our design), and $m$ is the kernel size (i.e., the number of weight parameters in a kernel). We hence can divide $a'$ into a set of regions evenly with the size of $\frac{W'}{z} \times \frac{H'}{z}$, denoted as $w \times h$. The feature representation $a'$ can hence be reshaped to $a' \in \mathbb{R}^{z \times z \times C' \times w \times h}$. The transformed feature representation $a''$ is obtained as follows.

$$a'' = \Psi(a') = \begin{bmatrix} r_{0,0} & r_{0,1} & ... & r_{0,z-1} \\ r_{1,0} & r_{1,1} & ... & r_{1,z-1} \\ ... & ... & ... & ... \\ r_{z-1,0} & r_{z-1,1} & ... & r_{z-1,z-1} \end{bmatrix},$$

$$r_{i,j} = \boldsymbol{U}[i,j] \otimes a'[i,j], \tag{2}$$

where $r_{i,j}$ denotes the transformed region $(i,j)$ and $\otimes$ denotes the convolutional operation. Observe that each region $a'[i,j]$ is transformed by a convolutional kernel $\boldsymbol{U}[i,j]$. These regions will be placed in their original positions after the transformation. Note that the transformed feature representation $a''$ has the same number of channels as $a'$ such that it can be properly decoded by the decoder to the input space. The variable $z$ for the number of regions is determined based on the size of the feature representation. In our current implementation, $z = \lceil \max(W', H')/32 \rceil + 2$. For example, assume an input whose feature representation size is $32 \times 32$, variable $z = \lceil 32/32 \rceil + 2 = 3$. We hence divide the feature representation into $3 \times 3$ regions. We also conduct a formal analysis in Appendix A to demonstrate that our regional transformation can express various backdoor behaviors.

**Loss Terms.** Figure 3 shows three loss terms. The SSIM score and the content loss are introduced to constrain the transformations on the inputs. In other words, it is desired to have generated backdoor samples retaining most main features and similar to the original inputs as backdoors typically preserve main contents (see Figure 1).

$$\mathcal{L}_{SSIM} = SSIM(x, \hat{x}), \quad \mathcal{L}_{content} = MSE(a, \hat{a}) = \frac{1}{N} \sum_{i=0}^{N-1} (a_i - \hat{a}_i)^2 \tag{3}$$

The cross entropy loss $\mathcal{L}_{CE}(y_t, \hat{y})$ is to induce the desired misclassification to the target label $y_t$.

Other than the above three loss functions, we also use another two loss terms to improve the quality of generated backdoor samples as follows.

$$\mathcal{L}_{norm} = \frac{1}{C} \sum^{C} |\mu_b - \bar{\mu}_{\boldsymbol{X}}| + \frac{1}{C} \sum^{C} |\sigma_b - \bar{\sigma}_{\boldsymbol{X}}|, \quad \mathcal{L}_{smooth} = MSE(\hat{x}, AvgPool(\hat{x})) \tag{4}$$

Loss term $\mathcal{L}_{norm}$ is to reduce the difference between the backdoor statistics (i.e., mean and standard deviation) and the average statistics across all the samples $\boldsymbol{X}$ (in the generation set) on each channel. This avoids the generated backdoor samples become too far away from the distribution of input samples. Loss term $\mathcal{L}_{smooth}$ smooths the local area of pixel changes, preventing abrupt pixel changes on the backdoor samples. Function $AvgPool$ is an average pooling operation, where each pixel value is replaced by the average of its neighboring pixels (e.g., in a $3 \times 3$ region).

Our final loss function for generating backdoors is the following.

$$\mathcal{L} = \mathcal{L}_{CE} + \alpha(\lambda_0 \mathcal{L}_{content} + \lambda_1 \mathcal{L}_{SSIM} + \lambda_2 \mathcal{L}_{smooth} + \mathcal{L}_{norm}) \tag{5}$$

We dynamically adjust the weight parameter $\alpha$ to balance the misclassification goal and the backdoor quality. We empirically set $\lambda_0 = 0.001$, $\lambda_1 = 100$, and $\lambda_2 = 0.05$ such that all the loss terms are at the same scale. The impact of these hyperparameters is studied in Appendix G.

## 5    EVALUATION

We evaluate MARTINI on three different computer vision tasks: image classification, self-supervised learning, and object detection. The defense performance of MARTINI is compared with 12 state-of-the-art backdoor mitigation techniques. We also carry out an adaptive attack to further test our approach and conduct an ablation study to understand the effects of different design choices. The extension of MARTINI to other domains is discussed in Appendix F.

### 5.1    MITIGATING BACKDOORS IN IMAGE CLASSIFIERS

**Experiment Setup.** We leverage a set of standard datasets and well-known model architectures. Five image classification datasets are employed in the experiments: CIFAR-10, STL-10, SVHN, GTSRB, and CelebA. Various model architectures such as ResNet, Network in Network (NiN), and VGG are used. We evaluate on **14 types of backdoor attacks** including BadNets (Gu et al., 2019), Dynamic attack (Salem et al.), Input-aware attack (IA) (Nguyen & Tran, 2020), DFST (Cheng et al., 2021), Blend attack (Chen et al., 2017), adaptive Blend attack (A-Blend) (Qi et al., 2023), Sinusoidal Signal attck (SIG) (Barni et al., 2019), LIRA (Doan et al., 2021), WaNet (Nguyen & Tran, 2021), Invisible attack (Li et al., 2021b), Clean Label attack (CL) (Turner et al., 2018), Narcissus (Zeng et al., 2023), COMBAT (Huynh et al., 2024), and filter attack (Liu et al., 2019). For filter attack, we make use of pre-trained models downloaded from the TrojAI competition (NIST). We consider **12 backdoor removal techniques**, including well-known defenses: Fine-tuning (FT) (Li et al., 2021a), Fine-pruning (FP) (Liu et al., 2018a), Mode Connectivity Repair (MCR) (Zhao et al., 2020), Neural Attention Distillation (NAD) (Li et al., 2021a), Adversarial Neuron Pruning (ANP) (Wu & Wang, 2021), Artificial Brain Stimulation (ABS) (Liu et al., 2019), Model Orthogonalization (MOTH) (Tao et al., 2022a); and recently proposed defenses: I-BAU (Zeng et al., 2021), SEAM (Zhu et al., 2023b), FT-SAM (Zhu et al., 2023a), FST (Min et al., 2024), RNP (Li et al., 2023). See details in Appendix B.

For evaluating the defense performance, the normal functionalities are measured using the predication accuracy on the test set (Acc.). We use the attack success rate (ASR) of backdoor attacks as the metric, which is the percentage of backdoor samples classified to the attack target label. We follow the same setup as in existing works (Li et al., 2021a; Tao et al., 2022a) by using only 5% of the original training set for mitigating backdoors.

**Comparison with Well-known Defenses.** Seven well-known backdoor mitigation approaches are used as baselines to compare with MARTINI. The defense results on the 14 backdoor attacks are presented in Table 1 and Appendix C. DFST (Cheng et al., 2021) introduces a detoxification procedure by iteratively training on reverse-engineered backdoors to reduce the number of compromised neurons that can be leveraged by existing defenses. We follow the original paper and evaluate two settings with one round (D1) and three rounds (D3) of detoxification[2].

The top three rows of Table 1 report the results on patch-type backdoors, where the trigger is a few pixel changes on the input, such as a square patch pattern. Almost all the defense techniques can reduce the ASR to less than 5%. MARTINI is able to reduce the ASR to less than 2% for all three attacks. These results are expected. As discussed in the motivation section, these static triggers can be easily learned by deep learning models, causing their learned features to be quite different from those of the main task. All the defenses can leverage this characteristic to easily isolate the backdoor behavior and eventually remove it.

DFST leverages a GAN to generate backdoor samples that are semantically similar to benign inputs. On CIFAR-10, FT, FP, MCR, NAD, ABS, and MOTH can only reduce the ASR of DFST on ResNet32-D1 from 97.60% to more than 60%. ANP is able to reduce the ASR to 20.67%, but at the cost of a significant accuracy degradation from 89.95% to 83.34%. MARTINI, on the other hand, can reduce the ASR to 14.22% with only a 1.73% accuracy degradation. NAD, ABS, and MOTH perform better on VGG13-D1, reducing the ASR from 95.89% to 24.56%, 33.78%, and 5.33%, respectively. However, with an increase in detoxification rounds, they can only reduce the ASR to more than 50%. MARTINI can consistently mitigate DFST backdoors, achieving less than 15% ASR on ResNet32 and less than 6% on VGG13.

The Blend attack uses random small perturbation patterns as the backdoor, which can be easily eliminated by all the evaluated techniques, except for FT on CIFAR-10. MARTINI can reduce the

---

[2]The original paper (Cheng et al., 2021) used at most three rounds of detoxification.

Table 1: Mitigating backdoors in image classifiers. The best results highlighted with blue color.

| Attack | Dataset | Model | Original | | FT | | FP | | MCR | | NAD | | ANP | | ABS | | MOTH | | MARTINI | |
|---|---|---|---|---|---|---|---|---|---|---|---|---|---|---|---|---|---|---|---|---|
| | | | Acc. | ASR | Acc. | ASR | Acc. | ASR | Acc. | ASR | Acc. | ASR | Acc. | ASR | Acc. | ASR | Acc. | ASR | Acc. | ASR |
| BadNets | CIFAR | ResNet18 | 93.50% | 100.0% | 90.88% | 1.26% | 91.58% | 1.16% | 91.51% | 1.10% | 92.57% | 0.87% | 90.72% | 4.07% | 92.28% | 0.40% | 89.47% | 1.11% | 88.49% | 1.97% |
| Dynamic | CIFAR | ResNet18 | 93.52% | 99.97% | 91.83% | 1.51% | 92.28% | 0.43% | 92.12% | 0.96% | 92.77% | 1.63% | 88.53% | 16.80% | 92.59% | 1.42% | 89.05% | 1.51% | 91.47% | 1.42% |
| IA | CIFAR | ResNet18 | 90.45% | 99.16% | 87.71% | 2.26% | 88.68% | 2.50% | 89.09% | 1.17% | 86.22% | 3.21% | 89.02% | 2.46% | 89.63% | 0.84% | 85.80% | 16.77% | 90.35% | 0.84% |
| DFST | CIFAR | RNet32-D1 | 89.95% | 97.60% | 87.30% | 70.00% | 88.20% | 62.89% | 87.95% | 62.67% | 88.00% | 60.44% | 83.34% | 20.67% | 86.97% | 84.11% | 88.90% | 65.44% | 88.22% | 14.22% |
| | | RNet32-D3 | 90.93% | 95.33% | 89.20% | 63.89% | 88.09% | 47.44% | 82.84% | 60.78% | 85.78% | 17.33% | 83.68% | 90.11% | 90.74% | 37.78% | 89.53% | 54.44% | 88.11% | 12.22% |
| | | VGG13-D1 | 90.34% | 95.89% | 86.07% | 93.56% | 87.37% | 51.11% | 86.58% | 90.33% | 87.24% | 24.56% | 86.92% | 89.56% | 88.71% | 33.78% | 87.26% | 5.33% | 88.03% | 2.00% |
| | | VGG13-D3 | 91.29% | 97.44% | 89.55% | 66.11% | 88.84% | 85.67% | 88.81% | 86.78% | 87.09% | 55.33% | 88.46% | 96.11% | 88.67% | 66.67% | 89.91% | 51.22% | 89.08% | 5.67% |
| | STL | RNet32-D1 | 75.74% | 97.67% | 70.64% | 70.64% | 68.89% | 96.11% | 68.05% | 84.67% | 70.92% | 44.00% | 65.71% | 90.11% | 72.26% | 68.56% | 71.97% | 60.89% | 72.10% | 2.67% |
| | | RNet32-D3 | 76.45% | 99.00% | 71.30% | 93.22% | 69.25% | 88.22% | 69.98% | 81.89% | 72.21% | 89.11% | 71.89% | 69.56% | 71.95% | 97.78% | 72.09% | 71.56% | 72.86% | 4.78% |
| | | VGG13-D1 | 72.18% | 98.67% | 70.11% | 84.78% | 67.12% | 67.00% | 66.06% | 66.22% | 68.91% | 86.67% | 68.88% | 98.11% | 68.46% | 48.44% | 69.86% | 62.67% | 68.61% | 5.89% |
| | | VGG13-D3 | 72.09% | 98.89% | 70.42% | 97.33% | 68.14% | 49.44% | 66.66% | 79.67% | 68.91% | 81.00% | 65.70% | 97.33% | 67.29% | 37.56% | 67.54% | 86.56% | 69.89% | 12.33% |
| Blend | CIFAR | ResNet20 | 90.96% | 99.96% | 90.33% | 84.92% | 87.75% | 3.63% | 85.53% | 63.58% | 86.81% | 3.94% | 85.20% | 6.22% | 89.41% | 5.66% | 85.44% | 12.26% | 89.08% | 0.00% |
| | SVHN | NiN | 94.10% | 92.37% | 92.70% | 0.54% | 88.26% | 23.75% | 93.50% | 0.59% | 94.40% | 0.33% | 92.67% | 0.41% | 91.76% | 10.66% | 94.41% | 0.20% | 94.56% | 0.85% |
| A-Blend | CIFAR | ResNet18 | 94.56% | 85.62% | 93.01% | 58.06% | 88.65% | 42.84% | 93.62% | 72.50% | 90.42% | 49.50% | 90.80% | 69.51% | 88.94% | 30.96% | 91.97% | 24.96% | 90.20% | 12.64% |
| SIG | CIFAR | ResNet20 | 83.38% | 93.30% | 88.65% | 59.40% | 81.01% | 76.29% | 83.31% | 16.63% | 85.84% | 9.44% | 80.02% | 37.44% | 86.82% | 85.29% | 80.39% | 17.72% | 86.91% | 3.97% |
| | SVHN | NiN | 95.48% | 92.46% | 95.11% | 41.60% | 93.35% | 23.49% | 93.19% | 45.16% | 94.21% | 0.68% | 90.10% | 20.91% | 93.47% | 55.02% | 94.90% | 0.69% | 93.96% | 0.46% |
| LIRA | CIFAR | ResNet18 | 93.25% | 99.92% | 91.19% | 23.60% | 90.60% | 77.53% | 92.24% | 27.22% | 90.61% | 11.55% | 88.58% | 48.17% | 91.12% | 7.01% | 90.06% | 37.90% | 91.78% | 6.59% |
| WaNet | CIFAR | ResNet18 | 94.15% | 99.55% | 93.58% | 80.71% | 89.14% | 2.09% | 93.29% | 1.74% | 91.37% | 0.87% | 91.38% | 0.11% | 89.61% | 1.88% | 92.65% | 0.62% | 91.12% | 0.64% |
| | GTSRB | ResNet18 | 99.01% | 98.94% | 96.80% | 48.75% | 96.06% | 63.40% | 98.54% | 10.47% | 94.96% | 0.02% | 97.38% | 0.00% | 98.51% | 0.00% | 97.72% | 0.01% | 97.70% | 0.30% |
| | CelebA | ResNet18 | 78.99% | 99.08% | 78.89% | 21.35% | 76.57% | 18.07% | 78.32% | 16.21% | 76.57% | 15.34% | 76.79% | 14.22% | 75.56% | 21.69% | 77.80% | 8.91% | 77.57% | 8.12% |
| Invisible | CIFAR | ResNet18 | 94.43% | 99.99% | 91.63% | 1.72% | 91.74% | 1.68% | 92.33% | 1.36% | 90.66% | 2.44% | 93.27% | 1.83% | 91.64% | 3.81% | 89.68% | 3.02% | 90.25% | 1.14% |
| | | VGG11 | 91.05% | 99.76% | 88.15% | 2.00% | 90.40% | 0.38% | 88.68% | 1.58% | 88.26% | 3.47% | 88.76% | 1.54% | 88.81% | 2.52% | 89.34% | 3.68% | 89.16% | 2.33% |
| CL | CIFAR | ResNet18 | 87.60% | 98.36% | 85.74% | 97.34% | 83.66% | 26.77% | 85.23% | 13.91% | 84.44% | 7.47% | 86.18% | 4.46% | 85.33% | 7.18% | 83.70% | 8.34% | 85.67% | 4.22% |
| Narcissus | CIFAR | ResNet18 | 92.38% | 94.76% | 90.74% | 52.83% | 90.18% | 69.99% | 92.09% | 76.25% | 88.29% | 52.05% | 89.79% | 92.06% | 88.52% | 34.20% | 90.49% | 50.31% | 90.51% | 1.45% |
| COMBAT | CIFAR | ResNet18 | 94.00% | 80.19% | 85.26% | 46.41% | 88.82% | 58.31% | 87.87% | 69.70% | 85.91% | 42.65% | 88.16% | 65.41% | 86.27% | 46.13% | 89.68% | 58.21% | 89.83% | 22.59% |
| **Average** | | | 88.79% | 96.56% | 86.67% | 50.55% | 85.39% | 41.61% | 85.90% | 41.33% | 85.73% | 26.56% | 84.88% | 41.49% | 86.21% | 31.57% | 85.98% | 28.17% | 86.62% | 5.17% |

ASRs to less than 1% for the two studied cases. However, when Blend is enhanced with an advanced attack strategy, A-Blend, which reduces the separation of clean and backdoored data distributions, existing defenses become less effective. Specifically, most of the baselines can only reduce the ASR to 40%. MOTH performs slightly better with a 25% ASR, as it does not rely on the separability of backdoor features. MARTINI can significantly reduce the ASR to 12.64%, surpassing all baselines.

The defense results on SIG, LIRA, WaNet, Invisible, and CL attacks are better for baselines, as existing techniques can reduce the ASRs to a reasonable range (less than 20% in most cases). MARTINI further reduces the ASRs to less than 5% for 7 out of 9 cases and less than 9% for the remaining cases (LIRA on CIFAR-10 and WaNet on CelebA), outperforming the well-known defenses.

Narcissus and COMBAT are recent backdoor attacks specifically designed to use features that closely resemble benign features. This design breaks the assumptions on which existing defenses are based, as discussed in the motivation section. As we can observe from the results in Table 1, almost all the baseline techniques cannot reduce the ASR to lower than 40%, except for ABS on Narcissus. In comparison, MARTINI can reduce the ASR of Narcissus to 1.45%, substantially lower than that achieved by the baselines. The ASR of COMBAT is also reduced to 22.59%, which is half that of existing techniques.

The results on filter attack are presented in Appendix C. MARTINI can eliminate all the backdoors with an average ASR down to 0.55%, outperforming the others. All the approaches incur a very small accuracy degradation on average ($< 0.3\%$).

**Comparison with Recent Defenses.** Five recent state-of-the-art backdoor mitigation methods are also utilized as baselines to compare with MARTINI. Table 2 reports the results. As these defense techniques have reported to be effective against several existing backdoor attacks, we hence focus on more recent advanced attacks: A-Blend, Narcissus, and COMBAT. These attacks proactively reduce the difference between backdoor features and benign features. According to Table 2, most of these defenses are effective against A-Blend, except for RNP. This is because RNP relies on backdoor-related neurons being more sensitive than benign neurons, similar to the assumption that ANP is based on. A-Blend is optimized to avoid such a characteristic, causing RNP to fail. For Narcissus and COMBAT, all the baseline defenses fall short, with ASRs remaining above 60% in almost all cases. FT-SAM and FST have slightly better results on Narcissus but are still less effective against COMBAT. Unlike A-Blend, which still uses an existing trigger pattern, Narcissus and COMBAT optimize the trigger such that not only are the model internals indistinguishable between backdoor and clean behavior, but the trigger pattern itself closely resembles benign features. Nevertheless,

Table 2: Comparison with recent defenses. The best results highlighted with blue color.

| Attack | Original | | I-BAU | | SEAM | | FT-SAM | | FST | | RNP | | MARTINI | |
|---|---|---|---|---|---|---|---|---|---|---|---|---|---|---|
| | Acc. | ASR | Acc. | ASR | Acc. | ASR | Acc. | ASR | Acc. | ASR | Acc. | ASR | Acc. | ASR |
| A-Blend | 94.56% | 85.62% | 90.89% | 25.01% | 90.70% | 14.60% | 88.93% | 20.76% | 90.74% | 10.60% | 88.01% | 52.69% | 90.20% | 12.67% |
| Narcissus | 92.38% | 94.76% | 90.37% | 61.21% | 88.80% | 69.07% | 87.82% | 30.02% | 86.86% | 41.49% | 88.62% | 72.01% | 90.51% | 1.54% |
| COMBAT | 94.00% | 80.19% | 90.22% | 63.28% | 89.79% | 72.00% | 89.75% | 67.33% | 91.66% | 73.79% | 89.20% | 75.94% | 89.83% | 22.59% |
| **Average** | **93.65%** | **86.86%** | **90.49%** | **49.83%** | **89.76%** | **51.89%** | **88.83%** | **39.37%** | **89.75%** | **41.96%** | **88.61%** | **66.88%** | **90.18%** | **12.27%** |

Table 3: Mitigating backdoors in self-supervised learning. The best results highlighted with blue.

| Attack | Pre-training Dataset | Downstream Task | Original | | FT | | NAD | | ANP | | Moth | | MARTINI | |
|---|---|---|---|---|---|---|---|---|---|---|---|---|---|---|
| | | | Acc. | ASR | Acc. | ASR | Acc. | ASR | Acc. | ASR | Acc. | ASR | Acc. | ASR |
| BadEncoder | CIFAR-10 | GTSRB | 82.43% | 99.37% | 77.13% | 92.35% | 77.75% | 91.52% | 83.31% | 28.30% | 75.97% | 52.28% | 82.58% | 5.10% |
| | | STL-10 | 76.26% | 99.79% | 74.85% | 97.59% | 74.67% | 92.88% | 73.20% | 32.90% | 72.60% | 88.99% | 72.53% | 13.84% |
| | | SVHN | 68.90% | 99.23% | 56.72% | 95.99% | 56.88% | 96.73% | 68.79% | 43.48% | 67.63% | 93.04% | 71.01% | 17.92% |
| | STL-10 | GTSRB | 74.25% | 98.46% | 61.05% | 45.45% | 64.38% | 3.88% | 70.48% | 4.51% | 68.88% | 46.39% | 73.52% | 3.90% |
| | | CIFAR-10 | 83.72% | 98.15% | 81.97% | 39.81% | 82.44% | 83.48% | 80.50% | 14.38% | 75.48% | 12.01% | 81.17% | 8.88% |
| | | SVHN | 74.36% | 96.28% | 66.06% | 21.15% | 65.81% | 25.58% | 66.38% | 23.25% | 69.16% | 27.28% | 71.72% | 14.95% |
| DRUPE | CIFAR-10 | GTSRB | 78.35% | 94.75% | 56.29% | 73.82% | 70.82% | 25.88% | 76.08% | 4.29% | 68.35% | 23.86% | 79.55% | 2.29% |
| | | STL-10 | 73.92% | 95.85% | 70.90% | 68.40% | 70.92% | 61.40% | 69.16% | 57.67% | 66.80% | 35.17% | 70.00% | 17.69% |
| | | SVHN | 79.40% | 95.59% | 67.60% | 3.97% | 67.64% | 79.54% | 79.25% | 89.03% | 74.01% | 4.49% | 74.20% | 0.68% |
| | STL-10 | GTSRB | 76.71% | 95.76% | 69.71% | 12.15% | 68.91% | 32.76% | 76.67% | 6.02% | 70.40% | 6.80% | 77.85% | 4.85% |
| | | CIFAR-10 | 84.14% | 95.31% | 84.51% | 14.19% | 84.87% | 22.24% | 81.63% | 19.00% | 82.19% | 16.89% | 83.71% | 6.98% |
| | | SVHN | 75.06% | 97.52% | 68.39% | 80.70% | 68.09% | 80.11% | 71.78% | 30.43% | 69.25% | 15.11% | 72.86% | 10.84% |
| **Average** | | | **77.29%** | **97.17%** | **69.60%** | **53.80%** | **71.10%** | **58.00%** | **74.77%** | **29.44%** | **71.73%** | **35.19%** | **75.89%** | **8.99%** |

MARTINI is still very effective against these attacks, achieving better defense performance compared to the most recent state-of-the-art defense techniques.

## 5.2 MITIGATING BACKDOORS IN SELF-SUPERVISED LEARNING

**Experiment Setup.** Self-supervised learning generates an encoder, used for downstream tasks. Backdoor attacks in this process involve adding a trigger to certain images in the training data. These altered samples are made to resemble specific target-class samples in the feature space. As a result, any input containing the trigger will be misclassified by a downstream classifier built on the compromised encoder. We leverage two backdoor attacks in self-supervised learning: BadEncoder (Jia et al., 2022) and DRUPE (Tao et al., 2023). The backdoor trigger is a 10x10 white square pattern. We use two datasets, CIFAR-10 and STL-10, as the pre-training datasets for constructing the encoder, and four datasets as the downstream tasks: GTSRB, SVHN, as well as CIFAR-10 and STL-10. We use ResNet18 and the contrastive learning algorithm SimCLR (Chen et al., 2020) for evaluation. Four baselines are adapted from image classification tasks: FT, NAD, ANP, and MOTH.

**Evaluation Results.** Table 3 reports the results. For BadEncoder, when CIFAR-10 is used as the pre-training dataset, most baseline techniques fail to remove the backdoor effects on the downstream tasks, with ASRs still over 80% in most cases. ANP performs better than other baselines, reducing the ASRs to around 30%. MARTINI achieves the best performance, reducing the ASR to as low as 5%. The observations on STL-10 are similar. For DRUPE, baseline defenses perform particularly poorly in certain cases. For example, when the pre-training dataset is STL-10 and the downstream is SVHN, both FT and NAD have ASRs over 80%. ANP has nearly 90% ASR on the CIFAR-10 encoder with SVHN as the downstream. Overall, MARTINI successfully mitigates backdoors in self-supervised learning, reducing the ASR from 97.17% to 8.99% with only a 1.4% accuracy drop.

## 5.3 MITIGATING BACKDOORS IN OBJECT DETECTION

**Experiment Setup.** We leverage the TrojAI (NIST) dataset for object detection. This dataset consists of images synthesized with real street backgrounds and multiple traffic signs as foreground objects. We consider three types of backdoor attacks (Chan et al., 2022; Chen et al., 2022; Lin et al.): misclassification, injection, and localization. Figure 5(a) shows the clean input and its cor-

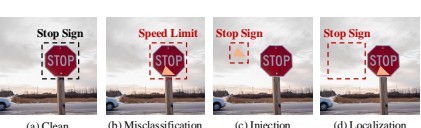

Figure 5: Object detection backdoors

rect prediction. Figure 5(b) demonstrates the object misclassification attack, where a yellow triangle, serving as the backdoor trigger, is stamped on the stop sign, causing the model to mis-recognize the sign as the speed limit. Figure 5(c) illustrates the injection attack, where the model falsely recognizes the trigger as a stop sign. Figure 5(d) visualizes the object localization attack, where the trigger causes the predicted object bounding box to shift away from its correct location. We conduct experiments using two well-known model architectures, SSD (Liu et al., 2016) and Faster-RCNN

(F-RCNN) (Ren et al., 2015). The clean object detection performance is measured by mean average precision (mAP). We adapte two baselines from image classification, i.e., FT and NAD.

**Evaluation Results.** The experimental results are presented in Table 4. On average, MARTINI reduces the ASR from 97% to below 6%, whereas the baselines still maintain ASR at high levels, over 34%, with greater performance sacrifices than MARTINI. This underscores the efficacy of MARTINI in mitigating backdoors in object detection, as it effectively approximates and eliminates the backdoor effects. Our mitigation is not perfect. In some cases, MARTINI still has a non-trivial ASR, such as misclassification and injection attacks

Table 4: Mitigating backdoors in object detection. The best results highlighted with blue color.

| Attack | Model | Original | | FT | | NAD | | Ours | |
|---|---|---|---|---|---|---|---|---|---|
| | | mAP | ASR | mAP | ASR | mAP | ASR | mAP | ASR |
| Miscls. | SSD | 87.74% | 100.00% | 81.85% | 92.38% | 80.60% | 75.24% | 82.82% | 9.05% |
| | F-RCNN | 96.50% | 100.00% | 92.13% | 71.43% | 91.23% | 15.24% | 92.95% | 5.71% |
| Inject. | SSD | 85.20% | 100.00% | 79.79% | 100.00% | 79.29% | 99.38% | 81.23% | 11.90% |
| | F-RCNN | 97.03% | 100.00% | 90.62% | 43.33% | 89.82% | 17.50% | 91.16% | 7.62% |
| Local. | SSD | 85.32% | 82.38% | 82.61% | 0.00% | 82.02% | 0.00% | 85.21% | 0.48% |
| | F-RCNN | 96.91% | 100.00% | 91.84% | 0.00% | 91.74% | 0.00% | 92.50% | 0.00% |
| **Average** | | **91.45%** | **97.06%** | **86.47%** | **51.19%** | **85.78%** | **34.56%** | **87.65%** | **5.79%** |

on SSD models. This might be due to the inherent robustness of backdoor attacks in object detection models, as these models involve complex predictions of bounding boxes and labels. Nevertheless, the results still demonstrate the generalizability of MARTINI to object detection.

### 5.4 OTHER EVALUATIONS

**Adaptive Attacks.** We conduct an adaptive attack by optimizing a trigger pattern during poisoning while applying our mitigation method (the adaptive knowledge). The repaired models by MARTINI all have less than 15% ASR, demonstrating the robustness of our technique (see Appendix D).

**Defense Efficiency.** We use an off-the-shelf encoder and only train the decoder, which takes 32.76 minutes. This is a one-time effort, and the trained decoder can be used for generating backdoors on different datasets. Our runtime efficiency is comparable to existing techniques (see Appendix E).

**Extension to Other Domains.** We extend MARTINI to NLP sentiment analysis. The results show that MARTINI can successfully elimiate backdoors, surpassing the baseline (see Appendix F).

**Ablation Study.** We study design choices of MARTINI individually to better understand their contributions. The four losses are all important. We also study the impact of four hyperparameters used in Equation 2 and Equation 5, and the impacts are small. See details in Appendix G.

## 6 RELATED WORK

In early works, backdoor attacks use a static trigger pattern, such as patch attacks (Gu et al., 2019; Chen et al., 2017). Recently, semantic backdoors have been explored by researchers. We have studied and evaluated several representative backdoor attacks in this paper, including DFST (Cheng et al., 2021), Blend attack (Chen et al., 2017), adaptive Blend attack (Qi et al., 2023), SIG (Barni et al., 2019), LIRA (Doan et al., 2021), WaNet (Nguyen & Tran, 2021), Invisible attack (Li et al., 2021b), Clean Label attack (Turner et al., 2018), Narcissus (Zeng et al., 2023), COMBAT (Huynh et al., 2024), and Filter attack (Liu et al., 2019).

Defense techniques against backdoor attacks can be categorized into backdoor input detection, certified robustness, backdoor scanning, and backdoor removal. *Backdoor input detection* aims to detect inputs stamped with backdoor triggers (Gao et al., 2019; Tran et al., 2018). *Certified robustness* provides certification to the classification results of individual samples, asserting the results can be trusted even in the presence of backdoors (McCoyd et al., 2020; Xiang et al., 2021a;b). *Backdoor scanning* focuses on identify whether a given model has been injected with backdoor (Kolouri et al., 2020; Tang et al., 2021). *Backdoor removal* aims to eliminate injected backdoors in poisoned models (Liu et al., 2018a; Zeng et al.). Our evaluation in Section 5.1 demonstrates the effectiveness of our method in mitigating backdoors, surpassing the 12 state-of-the-art techniques.

## 7 CONCLUSION

We propose a novel backdoor mitigation technique, MARTINI, that can eliminate a variety of backdoor attacks, including the most recent advanced attacks. It features a general backdoor generation method that models a spectrum of backdoors. The evaluation on various datasets and model architectures demonstrates that MARTINI can reduce the attack success rate of 14 backdoor attacks from 96.56% to 5.17%, outperforming 12 existing state-of-the-art defense techniques.

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

## A FORMAL ANALYSIS ON THE TRANSFORMATION LAYER

As a region of the feature representation is transformed by a convolutional kernel $U$, we study the property of such an operation for expressing backdoor behaviors. Assume a $2 \times 2$ input region $X$ on the left of Figure 6 and a kernel parameterized by $W \in \mathbb{R}^{2 \times 2}$. Zero-padding is used (demonstrated by the dotted cells). Output values can be derived from the values in the region and the parameter values through the following equations.

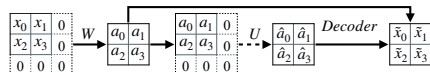

Figure 6: Example for transforming internal values

$$a_0 = w_0 \cdot x_0 + w_1 \cdot x_1 + w_2 \cdot x_2 + w_3 \cdot x_3$$
$$a_1 = w_0 \cdot x_1 + w_2 \cdot x_3$$
$$a_2 = w_0 \cdot x_2 + w_1 \cdot x_3$$
$$a_3 = w_0 \cdot x_3 \tag{6}$$

Here, we leave the activation functions out for discussion simplicity. Suppose a backdoor applies adversarial perturbation $\delta$ on the region $X$. That is, $x_i' = x_i + \delta_i, i \in \{0, 1, 2, 3\}$. The feature representation for the backdoor sample region $A'$ is hence the following.

$$a_0' = w_0 \cdot (x_0 + \delta_0) + w_1 \cdot (x_1 + \delta_1) + w_2 \cdot (x_2 + \delta_2)$$
$$\quad + w_3 \cdot (x_3 + \delta_3)$$
$$a_1' = w_0 \cdot (x_1 + \delta_1) + w_2 \cdot (x_3 + \delta_3)$$
$$a_2' = w_0 \cdot (x_2 + \delta_2) + w_1 \cdot (x_3 + \delta_3)$$
$$a_3' = w_0 \cdot (x_3 + \delta_3) \tag{7}$$

Our goal is to derive backdoor samples from benign inputs. That is, we apply the convolutional operation on the benign feature representation to produce the backdoor representation. Here, we use a convolutional kernel $U \in \mathbb{R}^{2 \times 2}$ for analysis simplicity. Applying the kernel on the normal representation $A$ (see the middle part of Figure 6) produces the following.

$$\hat{a}_0 = u_0 \cdot a_0 + u_1 \cdot a_1 + u_2 \cdot a_2 + u_3 \cdot a_3$$
$$\quad = u_0 \cdot (w_0 x_0 + w_1 x_1 + w_2 x_2 + w_3 x_3) + u_1 \cdot (w_0 x_1 + w_2 x_3)$$
$$\quad + u_2 \cdot (w_0 x_2 + w_1 x_3) + u_3 \cdot w_0 x_3$$
$$\hat{a}_1 = u_0 \cdot a_1 + u_2 \cdot a_3 = u_0 \cdot (w_0 x_1 + w_2 x_3) + u_2 \cdot w_0 x_3$$
$$\hat{a}_2 = u_0 \cdot a_2 + u_1 \cdot a_3 = u_0 \cdot (w_0 x_2 + w_1 x_3) + u_1 \cdot w_0 x_3$$
$$\hat{a}_3 = u_0 \cdot a_3 = u_0 \cdot w_0 x_3 \tag{8}$$

Let $A' = \hat{A}$ and we have

$$\delta_0 = (u_0 - 1) \cdot x_0 + u_1 \cdot x_1 + u_2 \cdot x_2 + u_3 \cdot x_3$$
$$\delta_1 = (u_0 - 1) \cdot x_1 + u_2 \cdot x_3$$
$$\delta_2 = (u_0 - 1) \cdot x_2 + u_1 \cdot x_3$$
$$\delta_3 = (u_0 - 1) \cdot x_3 \tag{9}$$

As observed in Figure 4, semantic backdoors transform inputs based on each original pixel value and do not introduce abrupt value changes in the neighborhood of each pixel (within the region)[3]. That is, each pixel perturbation introduced by the backdoor transformation correlates to the original value of its corresponding pixel and the neighboring pixels. This can be expressed by our method as show in Equation 9. For instance, the perturbation on the first pixel $\delta_0$ is a portion $(u_0 - 1)$ of the corresponding pixel $x_0$ and also the linear combination of neighboring pixels $(u_1 x_1 + u_2 x_2 + u_3 x_3)$. The scale of the perturbation is parameterized by our convolutional transformation $U$. It can be

---

[3]Static backdoors, such as BadNets, introduce strong features distinct from benign features. Our formulation can capture the introduced features, such as the color scheme. These features can be easily mitigated by MARTINI, as shown by our experiments in Section 5.1.

properly modeled during our backdoor generation using the gradient information from the victim model. Note that although $\delta_1$-$\delta_3$ may not involve some neighboring pixels, that is because we have only one layer. In practice, a model has many layers, and $x_0$-$x_3$ are feature values from the previous layer, which are functions involving neighboring pixels. In addition, the above analysis only considers one convolutional kernel in our transformation layer within the region for discussion simplicity. In practice, for example, the feature representation has 64 channels and each channel is associated with one kernel, which gives us 64 different combinations of neighboring pixels for each region.

## B  DETAILS OF EXPERIMENT SETUP

**Datasets and Models.**

- **CIFAR-10** (Krizhevsky et al., 2009) is an object recognition dataset with 10 classification classes. It consists of 60,000 images and is divided into a training set (48,000 images), a validation set (2,000 images), and a test set (10,000 images).

- **STL-10** (Coates et al., 2011) is an image recognition dataset with 10 classification classes. It consists of 5,000 training images and 8,000 test images.

- **SVHN** (Netzer et al.) is a dataset contains house digital numbers extracted from Google Street View images. It has 73,257 training images and 26,032 test images. We divide the original training set into 67,257 images for training and 6,000 images for validation.

- **GTSRB** (Stallkamp et al., 2012) is a German traffic sign recognition dataset with 43 classes. We split the dataset into a training set (35,289 images), a validation set (3,920 images), and a test set (12,630 images).

- **CelebA** (Liu et al., 2015) is a face attributes dataset. It contains 10,177 identities with 202,599 face images. Each image has an annotation of 40 binary attributes. We follow (Nguyen & Tran, 2021) to select 3 out of 40 attributes, i.e., Heavy Makeup, Mouth Slightly Open, and Smiling, and create an 8-class classification task.

- **TrojAI** (NIST) round 4 includes 16 types of model structures such as InceptionV3 (Szegedy et al.), DenseNet121 (Huang et al., 2017), SqueezeNet (Iandola et al.), etc. The task of these models is to recognize synthetic street traffic signs with between 15 and 45 classes. Input images are constructed by compositing a foreground object, e.g., a synthetic traffic sign, with a random background images from five different dataset such as Cityscapes (Cordts et al., 2016), KITTI (Geiger et al., 2013), Swedish Roads (Larsson et al., 2011), etc. A set of random transformations are applied during model training, such as blurring, lighting, shifting, titling, etc. Adversarial training such as PGD (Madry et al., 2018) and FBF (Wong et al., 2020) is also utilized to improve model quality. We randomly select 34 poisoned models by filter attack from TrojAI round 4 (NIST).

**Baselines**

- **Fine-tuning (FT)** (Li et al., 2021a) is a standard method originally proposed for transfer learning. It updates a pre-trained model's weights with a small learning rate on the training set. We leverage the finetuning baseline setting in NAD (Li et al., 2021a), which adopts data augmentation techniques including random crop, horizontal flipping, and cutout (DeVries & Taylor) during training.

- **Fine-pruning (FP)** (Liu et al., 2018a) prunes neurons that have low activation values for a set of clean samples. It then finetune the pruned model on a small set of clean samples.

- **MCR** (Zhao et al., 2020) linearly interpolates the weight parameters of two models. It also includes a set of trainable parameters during the interpolation. Specifically, the following equation is used to build a new model $\phi_\theta(t)$.

$$\phi_\theta(t) = (1-t)^2\omega_1 + 2t(1-t)\theta + t^2\omega_2, \quad 0 \le t \le 1, \tag{10}$$

where $t$ is the interpolation hyper-parameter ranging from 0 to 1. $\omega_1$ and $\omega_2$ are the weight parameters of two pre-trained models, which are fixed. $\theta$ is a set of trainable parameters that have the same shape of $\omega_1$ and $\omega_2$. For eliminating backdoors in poisoned models, MCR uses the poisoned model and its finetuned version as the two endpoints ($\omega_1$ and $\omega_2$)

and trains $\theta$ on a small set of clean samples. The best $t$ is chosen for the interpolation based on the clean accuracy.

- **NAD** (Li et al., 2021a) leverages the teacher-student structure to eliminate backdoors. It first finetunes the poisoned model on 5% of the training set. It uses this finetuned model as the teacher network, and the poisoned model as the student network. It then aims to reduce the internal feature differences between the teacher network and the student network by updating the student network. Finally, NAD outputs the student network as the cleaned model.

- **ANP** (Wu & Wang, 2021) is based on the observation that backdoor related neurons are more sensitive to adversarial perturbations on their weights. It hence applies a mask on all the neurons in the model, adversarially perturbs neuron weights to increase the classification loss for a set of clean samples, and minimizes the size of mask. ANP then prunes neurons with small mask values, meaning that they have been compromised by backdoor attacks.

- **ABS** (Liu et al., 2019) introduces a neuron stimulation analysis to expose abnormal behaviors of neurons in a deep neural network by increasing their activation values. Those neurons are regarded as compromised neurons and leveraged to reverse engineer backdoor triggers. ABS proposes a one-layer transformation to approximate/invert filter triggers. The inverted trigger is hence utilized to remove the injected backdoor in poisoned models following the unlearning procedure in NC (Wang et al., 2019).

- **MOTH** (Tao et al., 2022a) enhances model robustness by increasing the distance between classes. It employs trigger inversion techniques to generate adversarial samples that bridge class separations and utilizes asymmetric training to harden the model. MOTH mitigates backdoor effects by disrupting the shortcut connection between victim classes and the target class.

- **I-BAU** (Zeng et al., 2021) introduces a minimax formulation to mitigate the backdoor effect. Specifically, this method leverages the implicit hypergradient to address the interdependence between trigger synthesis and adversarial training processes.

- **SEAM** (Zhu et al., 2023b) leverages the phenomenon of catastrophic forgetting to unlearn the backdoor effect through label shuffling. It then seeks to restore clean knowledge by fine-tuning with the correct labels. This method effectively disrupts the connection between the backdoor trigger and its target label.

- **FT-SAM** (Zhu et al., 2023a) represents a novel backdoor defense paradigm that integrates sharpness-aware minimization with fine-tuning. This approach specifically targets neurons associated with the backdoor, aiming to reduce their influence by shrinking their norms, thereby mitigating the backdoor effect.

- **FST** (Min et al., 2024) proposes a simple yet effective technique for purifying backdoors through finetuning. It specifically promotes shifts in feature representation by actively diverging the classifier weights from their initially compromised states.

- **RNP** (Li et al., 2023) exposes and eliminates backdoor neurons through a process of unlearning followed by recovery. Specifically, RNP begins by maximizing the model's error using a small subset of clean data. Afterward, it recovers the affected neurons by minimizing the model's error on the same dataset. Neurons that remain problematic after this process are considered backdoored and are subsequently pruned.

**Backdoor Attacks**

- **BadNets** (Gu et al., 2019) is the pioneering study that first highlighted backdoor threats in deep learning models. It employs a static patch, e.g., a flower placed in the corner of an image, as the backdoor trigger. Any input containing this patch is then misclassified to the attack target label.

- **Dynamic attack** (Salem et al.) utilizes a generative model to create dynamic patch triggers that vary in pattern and location across images. This diversity enhances its stealthiness against backdoor detection methods.

- **Input-aware attack (IA)** (Nguyen & Tran, 2020) improves upon dynamic backdoors by generating more diverse and sample-specific triggers to evade backdoor detection. The IA triggers are more invisible in the input space.

- **Deep Feature Space Trojan (DFST)** (Cheng et al., 2021) leverages a generative adversarial network (GAN) to inject a certain style (e.g., sunrise color style) to given training samples. It also introduces a detoxification procedure by iteratively training on ABS (Liu et al., 2019) reverse-engineered backdoors to reduce the number of compromised neurons that can be leveraged by existing scanners for successful detection. We follow the original paper and poison models with two settings: one-round detoxification and three-rounds detoxification.

- **Blend attack** (Chen et al., 2017) injects a random perturbation pattern on the training samples of non-target classes and changes the ground truth labels of these samples to the target class (label 0). We use the random pattern reported in the original paper and use a blend ratio of $\alpha = 0.2$.

- **Adaptive Blend (A-Blend)** (Qi et al., 2023) refines the typical blend attack by including correctly labeled trigger-planted samples to enhance backdoor learning regularization. It also introduces asymmetric trigger strategies that improve the ASR and diversify the representations of poisoned samples.

- **Sinusoidal Signal attck (SIG)** (Barni et al., 2019) injects a strip-like pattern on the training samples of the target class and retains the original ground truth labels. We follow the setting in the original paper and generate the backdoor pattern using the horizontal sinusoidal function with $\Delta = 20$ and $f = 6$. We use label 0 as the target class and poison 8% of the training data in the target class.

- **LIRA** (Doan et al., 2021) designs a learnable trigger injection function to be used during model poisoning. Specifically, it trains a generative model to inject triggers concurrently with backdoor model training. LIRA utilizes the dataset itself to enhance the specificity of the backdoor triggers.

- **WaNet** (Nguyen & Tran, 2021) uses elastic image warping that deforms an image by applying the distortion transformation (e.g., distorting straight lines) as the backdoor. We download three backdoored models from the official repository (Nguyen & Tran, 2021), which are trained on CIFAR-10, GTSRB, and CelebA, respectively.

- **Invisible attack** (Li et al., 2021b) leverages a generator to encode a string (e.g., the index of a target label) onto an input image. We download the pre-trained generator from the official repository (Li et al.) and use it to inject invisible backdoors following the setting in the original paper.

- **Clean Label attack (CL)** (Turner et al., 2018) generates adversarial perturbations on the training samples in the target class using an adversaraily trained model. It then injects a $2 \times 2$ grid at the top left corner of the target-class inputs and retain their ground truth labels. We use $L^\infty$ bound of $8/255$ for crafting adversarial perturbations, use label 3 as the target class, and poison 50% of the training data in the target class following the official repository (Turner et al.).

- **Narcissus** (Zeng et al., 2023) introduces a clean-label backdoor attack that is both stealthy and robust. Specifically, it trains a surrogate model to capture the important features from the target label, which are then used as the backdoor trigger. It selectively poisons only the images of the target class with this trigger, compelling the model to associate the trigger with the target label without altering the labels.

- **COMBAT** (Huynh et al., 2024) improves the clean-label backdoor technique beyond Narcissus by leveraging a generative model to produce the triggers. It also incorporates frequency features and introduces an alternative training method to enhace the learning of the backdoor trigger function and the poisoned model.

- **Filter attack** (Liu et al., 2019) applies Instagram filters on training samples and changes the ground truth labels of these samples to the target class. There are various filters can be used to poison data, such as Gotham filter, Nashville filter, Kelvin filter, Lomo filter, Toaster filter, etc.

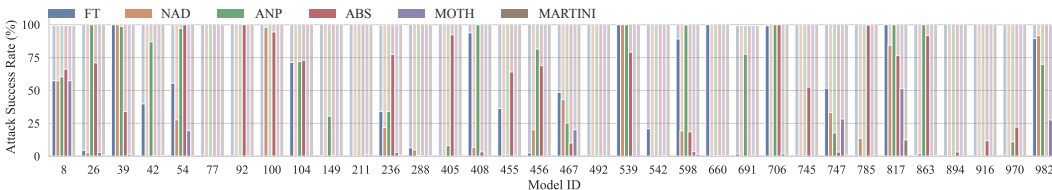

Figure 7: ASR of filter attack before (light color) and after (dark color) applying different defenses

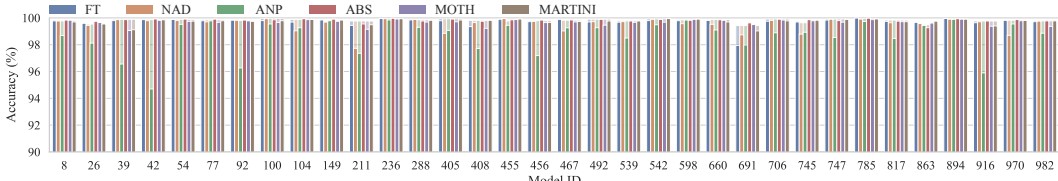

Figure 8: Acc. of filter-backdoored models before (light color) and after (dark color) applying different defenses

## C RESULTS ON FILTER ATTACK

For filter attack, we leverage pre-trained backdoored models downloaded from the TrojAI competition (NIST). The results are reported in Figure 7. The x-axis and y-axis denote the model IDs and the ASR, respectively. The bars in the light/dark colors show the ASR of injected backdoors before/after applying different defense techniques. Observe that FT (blue bars) can only repair half of the evaluated models (17 out of the 34 models). This is expected as backdoor attacks include clean data together with backdoored samples during training. Fine-tuning only on clean samples may not eliminate the backdoor patterns that have already been learned by backdoored models. NAD leverages the teacher-student structure and treats the model from FT as the teacher network. Its performance hence is limited by Fine-tuning. This can be observed from the orange bars in Figure 7. NAD is only able to eliminate five more backdoors (with a total of 21 models). ANP has limited performance on TrojAI models, with only 15 poisoned models being repaired. The TrojAI backdoored models were trained by NIST (NIST), and different training strategies including random transformations, adversarial training, etc., were employed to make injected backdoors more robust and hard to detect. These strategies may reduce the sensitivity of individual neurons on backdoor patterns. ANP is hence not able to identify compromised neurons and fails to remove injected backdoors. This observation is consistent with the results on DFST backdoors that apply detoxification to reduce compromised neurons.

ABS can only repair 15 models. As the injected backdoors in TrojAI models are label-specific, ABS may not be able to identify the correct victim-target class pair. The inverted triggers fail to expose the injected backdoor behaviors. Unlearning on those triggers hence cannot repair models. MOTH can eliminate more backdoors than other baselines with 28 fixed models. As discussed in the motivation section, semantic backdoors perturb all pixels on the input and are dynamic, while MOTH focuses on patch-like static backdoors. It can raise the bar for semantic backdoors to some extent but still fails to repair 6 TrojAI backdoored models. MARTINI, on the other hand, can eliminate all the backdoors with an average ASR down to 0.55%, outperforming the others. The accuracy of backdoored models before and after repair is shown in Figure 8 (in Appendix). Overall, all the approaches incur a very small accuracy degradation on average ($< 0.3\%$), except for ANP (1.16%). MARTINI has the smallest accuracy degradation of 0.06%.

## D ADAPTIVE ATTACKS

We conduct an adaptive attack by optimizing a trigger pattern during poisoning while applying our mitigation method (the adaptive knowledge). The goal is to prevent the model from learning simple triggers that can be easily generated, making the final injected backdoor hard to invert and capable of evading our defense. The adaptive attack starts with a random trigger and stamps it on training images along with the target label for poisoning. At each training iteration, it also applies the inverted trigger, stamps it on images, and uses the ground truth label for training. The attack then optimizes

the injected trigger on the current adversarially-trained model and uses this optimized trigger for injection. The poisoning process is iterative and continues until convergence.

The experiment is conducted on a ResNet20 model with CIFAR-10, evaluating different choices of the 10 classes as the target label. The results are shown in Table 5. The first column shows the target label. Columns 2-5 show the accuracy and ASR for the backdoored models (with a 20% poisoning rate) before and after re-pair by our method. Columns 6-9 present the results using a 50% poisoning rate for adaptive attacks. Observe that with a 20% poisoning rate, the backdoored models have an average accuracy of 83.12% and an ASR of 66.20%. By increasing the poisoning rate to 50%, the ASR

Table 5: Adaptive Attacks

| Target | 20% Poisoning | | MARTINI | | 50% Poisoning | | MARTINI | |
|---|---|---|---|---|---|---|---|---|
| | Acc. | ASR | Acc. | ASR | Acc. | ASR | Acc. | ASR |
| 0 | 83.64% | 60.43% | 83.94% | 9.94% | 67.86% | 74.94% | 75.71% | 4.42% |
| 1 | 81.61% | 58.48% | 84.57% | 10.46% | 70.64% | 79.99% | 81.79% | 6.01% |
| 2 | 84.43% | 56.88% | 83.67% | 12.79% | 67.99% | 77.74% | 81.50% | 5.70% |
| 3 | 84.91% | 60.22% | 85.56% | 7.18% | 70.73% | 74.81% | 75.02% | 0.79% |
| 4 | 82.91% | 76.90% | 83.29% | 10.55% | 71.87% | 75.35% | 75.80% | 9.47% |
| 5 | 80.11% | 76.04% | 84.87% | 5.00% | 66.70% | 78.09% | 79.12% | 11.28% |
| 6 | 84.47% | 64.13% | 82.78% | 12.93% | 71.93% | 73.24% | 84.72% | 14.31% |
| 7 | 82.32% | 76.69% | 83.71% | 13.11% | 68.37% | 84.39% | 78.61% | 11.09% |
| 8 | 84.18% | 62.56% | 86.06% | 12.16% | 71.50% | 75.27% | 82.92% | 12.86% |
| 9 | 82.57% | 69.63% | 84.63% | 9.14% | 70.85% | 75.50% | 79.20% | 10.29% |

improves to 76.93%, with a significant accuracy degradation to 69.84% on average. The ASRs are slightly higher for target labels 4, 5, and 7 with the 20% poisoning rate, and for label 7 with the 50% poisoning rate. As MARTINI aims to mitigate backdoors while the poisoning tries to inject a backdoor, these two contradicting goals result in the accuracy being much lower than a clean model (91.52%) and the ASR being relatively low as well. By applying our method to the poisoned models, the ASR drops to 10.33% (20% poisoning rate) and 8.62% (50% poisoning rate) without accuracy degradation (84.31% and 79.44% on average, respectively), as shown in Table 5. This delineates the resilience of our mitigation technique to adaptive attacks. Regarding different target labels, the ASRs for repaired models are slightly higher for labels 2, 6, and 7 with the 20% poisoning rate, and for labels 6 and 8 with the 50% poisoning rate. These slight variations are due to the fact that our method does not mitigate backdoors equally for all classes. Nonetheless, the repaired models all have less than 15% ASR, demonstrating the robustness of our technique against adaptive attacks with different target choices.

We also conduct an adaptive attack where backdoors have the same feature in different regions to counter our regional transformation. The results show that our method reduces the ASR from 98.80% to 0.28% with only a 0.83% accuracy degradation.

## E DEFENSE EFFICIENCY

We use an off-the-shelf encoder and only train the decoder, which takes 32.76 minutes. This is a one-time effort, and the trained decoder can be used for generating backdoors on different datasets.

Figure 9 shows the time in seconds for mitigating backdoors by different defenses. Most of the defense techniques can finish within 100 seconds. MCR, ABS, and MOTH have higher time costs, requiring more than 250 seconds. Overall, the time cost of MARTINI is comparable to that of other baselines. Recall that our method achieves more than 20% ASR reduction compared to baselines on many attacks, especially on recent advanced attacks, which is a critical aspect of backdoor defense.

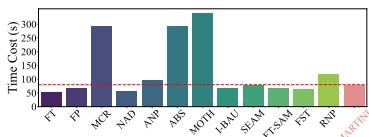

Figure 9: Time Cost

## F EXTENSION TO OTHER DOMAINS

The idea of our technique can be extended to defend against natural language processing (NLP) backdoors by leveraging a sentence-to-sentence model. A specially designed transformation layer can transform abstract features of sentences to embed backdoor effects. Adversarially training on the generated backdoor samples can then eliminate these NLP backdoors.

We apply MARTINI to NLP sentiment analysis. We use a pretrained DistilBERT as the generator, insert our transformation

Table 6: Mitigating backdoors in NLP

| Poisoned Model | Original | | NAD | | Ours | |
|---|---|---|---|---|---|---|
| | Acc. | ASR | Acc. | ASR | Acc. | ASR |
| Model-1 | 85% | 92% | 85% | 29% | 84% | 18% |
| Model-2 | 89% | 94% | 88% | 27% | 88% | 17% |
| Model-3 | 87% | 91% | 88% | 29% | 87% | 14% |

layer before the decoding layer, and adversarially train the model.

We leverage three TrojAI-round5 poisoned models (injected with a phrase trigger), and the results are reported in Table 6. We adapt NAD from image classification to this setting. From the table, we observe that `MARTINI` reduces ASR to 16% on average, with a 1% accuracy degradation. The baseline NAD can only reduce ASR to 28%. This result shows that `MARTINI` has good potential for defending NLP backdoors. We leave further exploration to future work.

# G    ABLATION STUDY

`MARTINI` features a few important design choices. In this section, we aim to study these choices individually to better understand their contributions to the performance. In particular, we study the effects of four loss terms used in backdoor generation. The ablation study is conducted on a ResNet20 model with CIFAR-10, and the results are presented in Table 7. Row 1 denotes the original backdoored model and row 2 the final result of our method. Rows 3-6 present the results of excluding each loss term individually during backdoor mitigation.

Observe that $\mathcal{L}_{content}$ can boost the performance by 37.45%. This is because it constrains the difference of feature representations between backdoor samples and normal inputs. Without it, the backdoor samples can be too different from normal inputs internally and the model cannot learn the correct features. $\mathcal{L}_{SSIM}$ and $\mathcal{L}_{smooth}$ have similar ASR reduction. The SSIM score directly constrains the quality of generated backdoor samples looking similar to original inputs. $\mathcal{L}_{smooth}$ further smooths the backdoor samples to improve the generated quality. $\mathcal{L}_{norm}$ on the normalization layer is also quite important as it makes sure the normalized inputs are not far from the original distribution.

Table 7: Ablation study on different design choices

| Method | Accuracy | ASR |
|---|---|---|
| Original | 91.52% | 81.36% |
| MARTINI | 90.31% | 1.41% |
| w/o $\mathcal{L}_{content}$ | 90.13% | 38.86% |
| w/o $\mathcal{L}_{SSIM}$ | 90.36% | 35.93% |
| w/o $\mathcal{L}_{norm}$ | 90.27% | 43.44% |
| w/o $\mathcal{L}_{smooth}$ | 90.13% | 32.04% |

**Impact of Hyperparameters.** We study the impact of four hyperparameters used in Equation 2 and Equation 5. The study is conducted on a backdoored model by DFST on STL-10, which has 72.18% accuracy and 98.67% ASR. The results are shown in Table 8. Observe that the impacts are small. Most settings can achieve good ASR reduction. In comparison, the lowest ASR achieved by the baselines is 48.44%. The best λs are chosen based on that all the loss terms are at the same scale as discussed below Equation 5. That is, the weighted loss value for each term shall be similar.

Table 8: Impact of hyperparameters

| z | 1 | 2 | 3 | 4 |
|---|---|---|---|---|
| Accuracy | 69.26% | 67.99% | **68.61%** | 67.66% |
| ASR | 17.00% | 8.33% | **5.89%** | 10.44% |
| $\lambda_0$ | 0.0005 | 0.001 | 0.002 | 0.005 |
| Accuracy | 68.59% | **68.61%** | 67.49% | 68.24% |
| ASR | 16.56% | **5.89%** | 11.11% | 13.67% |
| $\lambda_1$ | 50 | 100 | 150 | 200 |
| Accuracy | 68.96% | **68.61%** | 68.42% | 68.49% |
| ASR | 15.11% | **5.89%** | 12.33% | 9.33% |
| $\lambda_2$ | 0.03 | 0.05 | 0.1 | 0.2 |
| Accuracy | 68.76% | **68.61%** | 68.44% | 68.34% |
| ASR | 9.44% | **5.89%** | 6.67% | 14.89% |

