# OpenReview forum: "Combating Hidden Vulnerabilities in Computer Vision Tasks"
_ICLR.cc/2025/Conference — ICLR 2025 Conference Withdrawn Submission_

### Official Review · Reviewer_tW9M · 2024-10-23

**Soundness:** 3
**Presentation:** 4
**Contribution:** 3
**Rating:** 5
**Confidence:** 4

**Summary:**

This paper proposes MARTINI, a new backdoor mitigation technique that effectively counters various backdoor attacks in deep learning models, including semantic-based triggers. MARTINI uses a trigger reverse-engineering method to generate backdoor samples with a similar attack effect, which, when paired with correct labels in training, removes backdoor effects.

**Strengths:**

1. The experiments are highly detailed, covering a wide range of tasks from image classification to object detection.

2. The paper provides a thorough and detailed explanation of the methodology, making it easy for me to fully grasp the nuances and specifics of the approach.

3. The method appears to be effective from an intuitive standpoint.

**Weaknesses:**

1. The primary issue is an unclear threat model. In the Trigger Reverse-engineering section, how does the defender know the attacker's target class? Additionally, how does the proposed method handle an ALL-to-ALL attack, and a source specific attack?

2. The proposed approach involves numerous modules and loss functions, which may make it difficult for the community to reproduce the results and fully utilize the method. While the authors provide justification for the necessity of the four loss functions in the ablation study in the appendix, I still have concerns. Why does it seem that each component contributes equally to the overall result? For instance, the performance degrades significantly when $L_{smooth}$  is removed. Although I understand its value for handling more stealthy backdoor attacks, why does it also perform well with an obvious trigger, especially in tasks like object detection with conspicuous triggers? My perspective is that for a method to be truly effective and broadly applicable, there must be a key component that plays a primary role, rather than having every part constantly contribute equally to the outcome.

3. The authors claim that the proposed approach can mitigate backdoors in self-supervised learning and object detection. I am concerned this might be an overclaim. Firstly, the authors do not compare their method with existing backdoor defense approaches for object detection tasks (such as Django[1] and ODSCAN[2]). My main concern is that triggers in object detection tasks are often more obvious patches, so how does the proposed method effectively reverse-engineer these trigger patterns? As I mentioned in my second point, this may conflict with components such as $L_{smooth}$ and other modules in the approach. I would like to see examples of the reverse-engineered triggers in object detection tasks, especially for classic trigger patterns like "black-and-white checkerboard," "random," or "watermelon."



[1] Django: Detecting Trojans in Object Detection Models via Gaussian Focus Calibration, NeurIPS 2023.
[2] ODSCAN: Backdoor Scanning for Object Detection Models, S&P 2024.

**Questions:**

I would like the authors to provide explanations for the concerns raised in the first and second points. Additionally, for the third point, I would like to see visual examples of the reverse-engineered triggers in object detection tasks, particularly for classic trigger patterns such as "black-and-white checkerboard," "random," or "watermelon". If I had a clearer understanding of the threat model, I believe I would have given a higher score.

---

### Official Review · Reviewer_dmLk · 2024-10-29

**Soundness:** 2
**Presentation:** 2
**Contribution:** 2
**Rating:** 5
**Confidence:** 4

**Summary:**

This paper proposes a trigger reverse-engineering method (MARTINI) for backdoor defense, which optimizes a transformation layer to induce the backdoor behavior on clean samples for trigger reconstruction. Extensive experiments are conducted with various backdoor attack & defense methods on five datasets to evaluate the effectiveness of MARTINI.

**Strengths:**

1) A new backdoor defense method has been proposed for mitigating the backdoor threat.

2) Extensive experiments are conducted on five datasets to demonstrate the effectiveness.

**Weaknesses:**

1) . The authors' intuition behind the MARTINI is that the perturbation of backdoor triggers is dependent on the original image pixel values in its neighboring area. This assumption is inconsistent with the facts that many attackers inject the invisible sample-agonistic trigger to induce backdoor attacks [1,2] or more complex triggers [3]. Although the experiment seems demonstrate the effectiveness of the MARTINI on those independent cases ( BadNets, SIG), the theoretical analysis is needed.

2) In fact, trigger reverse-engineering methods hardly reconstruct the original injected triggers instead of the same effect adversarial noise (like NC on most cases). How to only employ a simple transformation layer to ensure the high-quality reconstruction in Figure 1? And according to the pipeline shown in Figure 3 and the following loss function, it's confused to generate such vivid triggers from the clean samples.

3) Is $a$ different from $a'$? Figure 1 is discussed in lines of 143 and 164 with different backdoor defense methods, where the figure needs to be refined for better readability.

4) Writing typos should be revised, such line 37.

5) Motivation section is not clear to support employing the transformation layer to achieve backdoor defense.

6) Implementation details about MARTINI are not reported.


[1] Liu Y, Ma X, Bailey J, et al. Reflection backdoor: A natural backdoor attack on deep neural networks[C]//Computer Vision–ECCV 2020: 16th European Conference, Glasgow, UK, August 23–28, 2020, Proceedings, Part X 16. Springer International Publishing, 2020: 182-199.
[2] Wang R, Wan R, Guo Z, et al. Spy-Watermark: Robust Invisible Watermarking for Backdoor Attack[J]. arXiv preprint arXiv:2401.02031, 2024.

[3]Zhang J, Dongdong C, Huang Q, et al. Poison ink: Robust and invisible backdoor attack[J]. IEEE Transactions on Image Processing, 2022, 31: 5691-5705.

**Questions:**

How do you pair the decoded image with the target label? Like NC?

---

### Official Review · Reviewer_UraD · 2024-11-03

**Soundness:** 3
**Presentation:** 3
**Contribution:** 3
**Rating:** 5
**Confidence:** 5

**Summary:**

This paper presents MARTINI, a mitigation technique that removes backdoor from models. It takes a general approach by modeling and reverse-engineering backdoor triggers through feature transformation. The method uses a specially designed transformation layer that can approximate various types of backdoor attacks by refining feature vector. It then generates synthetic backdoor samples using this transformation layer and uses them along with correct labels to train out the planted backdoor. They also show MARTINI's generalizability to self-supervised learning and object detection tasks.

**Strengths:**

It shows good performance in defending against backdoor attacks with complex and semantic triggers.

The design is trivial, easy to understand, and can be applied to classification, self-supervised learning, and object detection.

**Weaknesses:**

The autoencoder-based trigger reverse engineering has been around since 2020 (e.g., Gangsweep, De-trigger autoencoder, etc.), and it makes good sense to model and reconstruct complex triggers in the feature space rather than the pixel domain. However, the performance of this scheme is constrained by the quality and effectiveness of the autoencoder, as it must be jointly trained (either fully or partially) using feedback from the classifier. This increases the difficulty of training by effectively raising the complexity of the model and objective function, especially for complex datasets with high-resolution inputs. The experimental results only show images with a maximum resolution of 178*218, which raises concerns about its performance on larger datasets.

Another concern is that it appears a separate transformation layer must be trained for each possible target class to reconstruct the feature space trigger. This would significantly increase the computational overhead for larger datasets with more classes, such as ImageNet, which has 1,000 classes.

Since the proposed scheme lacks a detection mechanism, it will always require extensive autoencoder training, trigger reconstruction, and classifier fine-tuning to remove a hypothetical existing backdoor. This introduces unnecessary training overhead and could lead to performance degradation or distribution drift, as the new dataset with the recovered trigger might have a slightly different distribution. This makes the approach impractical for many real-world applications.

**Questions:**

What is the performance of the proposed scheme on larger datasets?

Could you discuss the computational overhead and the potential impact on the classifier? Please refer to the weaknesses for a detailed explanation.

---

### Official Review · Reviewer_F6fM · 2024-11-04

**Soundness:** 2
**Presentation:** 3
**Contribution:** 2
**Rating:** 5
**Confidence:** 3

**Summary:**

This paper presents MARTINI, an innovative backdoor defense mechanism designed to effectively mitigate a diverse range of backdoor attacks in deep learning models. MARTINI operates by reverse-engineering trigger methods to generate transformed samples that resemble backdoor-affected inputs. These samples are subsequently used to retrain the model, aiming to weaken or eliminate backdoor vulnerabilities. The method is evaluated across multiple tasks, including self-supervised learning and object detection, demonstrating its capability to significantly reduce attack success rates with minimal impact on model accuracy. MARTINI proves to be versatile, adaptable to various types of backdoors, and resilient across different attack scenarios, providing a robust and comprehensive defense against backdoor threats in deep learning models.

**Strengths:**

This paper introduces a trigger reverse-engineering mechanism and a novel transformation layer that enables it to generalize across various backdoor attacks, including those that modify abstract features.

This paper provides extensive evaluation on 14 types of backdoor attacks across several datasets and model architectures and can outperform 12 state-of-the-art techniques by significantly reducing the ASR within an acceptable process time.

**Weaknesses:**

MARTINI's search process primarily optimizes a single type of backdoor attack and generates diverse trigger patterns within that type by adjusting the transformation layer parameters. However, when faced with multi-trigger or complex backdoor attacks, this simple iterative search may struggle to effectively identify the optimal parameter combinations, potentially leading to limitations in MARTINI's effectiveness in detecting and defending against such advanced backdoor attacks.

Some critical information is missing in the evaluation section. For example, what are the hardware settings used for evaluation? What are the training parameters used for the decoder and victim model during model training? Without such information, MARTINI’s effectiveness may be limited, especially in resource-constrained or high real-time demand scenarios.


The paper primarily evaluates MARTINI on small and medium-sized deep learning models and datasets; however, its performance and efficiency on large-scale pretrained models (such as GPT-3, LLaMA-7B, etc.) remain unclear.

**Questions:**

While MARTINI shows promising results on small and medium-sized models, is it scalable to large-scale models, such as LLaMA-7B or GPT-3? Understanding potential challenges or limitations in applying MARTINI to these larger models would clarify its practical applicability in real-world, large-scale settings.

---

### Note · Authors · 2024-11-14

I have read and agree with the venue's withdrawal policy on behalf of myself and my co-authors.